# Climate and Land-Cover Change Impacts and Extinction Risk Assessment of Rare and Threatened Endemic Taxa of Chelmos-Vouraikos National Park (Peloponnese, Greece)

**DOI:** 10.3390/plants11243548

**Published:** 2022-12-15

**Authors:** Konstantinos Kougioumoutzis, Panayiotis Trigas, Maria Tsakiri, Ioannis P. Kokkoris, Eleni Koumoutsou, Panayotis Dimopoulos, Dimitris Tzanoudakis, Gregoris Iatrou, Maria Panitsa

**Affiliations:** 1Laboratory of Botany, Department of Biology, University of Patras, 26504 Patras, Greece; 2Laboratory of Systematic Botany, Department of Crop Science, Agricultural University of Athens, 11855 Athens, Greece

**Keywords:** *Alchemilla aroanica*, *Silene conglomeratica*, *Lonicera alpigena* subsp. *hellenica*, *Polygala subuniflora*, *Valeriana crinii* subsp. *crinii*, *Corydalis blanda* subsp. *oxelmannii*, *Globularia stygia*, IUCN criteria, climate change scenarios, integrated protected area management

## Abstract

Chelmos-Vouraikos National Park is a floristic diversity and endemism hotspot in Greece and one of the main areas where Greek endemic taxa, preliminary assessed as critically endangered and threatened under the IUCN Criteria A and B, are mainly concentrated. The climate and land-cover change impacts on rare and endemic species distributions is more prominent in regional biodiversity hotspots. The main aims of the current study were: (a) to investigate how climate and land-cover change may alter the distribution of four single mountain endemics and three very rare Peloponnesian endemic taxa of the National Park via a species distribution modelling approach, and (b) to estimate the current and future extinction risk of the aforementioned taxa based on the IUCN Criteria A and B, in order to investigate the need for designing an effective plant micro-reserve network and to support decision making on spatial planning efforts and conservation research for a sustainable, integrated management. Most of the taxa analyzed are expected to continue to be considered as critically endangered based on both Criteria A and B under all land-cover/land-use scenarios, GCM/RCP and time-period combinations, while two, namely *Alchemilla aroanica* and *Silene conglomeratica,* are projected to become extinct in most future climate change scenarios. When land-cover/land-use data were included in the analyses, these negative effects were less pronounced. However, *Silene conglomeratica*, the rarest mountain endemic found in the study area, is still expected to face substantial range decline. Our results highlight the urgent need for the establishment of micro-reserves for these taxa.

## 1. Introduction

Climate has been changing since the 1800s mostly due to human activities, with global mean annual temperature expected to reach or exceed 1.5 °C of warming averaged over the next 20 years bringing multiple changes in many areas of the world concerning mainly wetness and dryness, winds, snow, and ice [1]. Over the last 200 years, the combined effects of climate and land-use change have led to species extinctions, biodiversity loss and homogenization all over the world (among others, [2,3,4,5,6,7,8]), with land-use change ranked as the greatest threat to nature and biodiversity by [9]. Moreover, both of these threats [i.e., climate change and land use/land cover (LULC) change] may act in tandem and amplify the negative effects of each other on biodiversity [10].

The Mediterranean Basin, a global biodiversity hotspot exhibiting high proportions of endemism mostly in insular and montane regions [2,11] is unfortunately also considered as an extinction hotspot due to human-induced climate change [12,13,14,15], showing relatively high levels of anticipated loss of wilderness areas in the coming decades, even within protected areas [16]. Extinction risk assessments of regional endemic and rare species across different climate change scenarios are essential for their conservation and management planning strategy [17,18,19,20]. Conservation strategies are needed to address the ongoing loss of plant diversity, and especially for the protection of rare and threatened plant species in the Mediterranean Basin [21].

Regional level studies concerning local endemic and rare species extinction risk, population status, and distributions under different climate change scenarios are critical for the conservation and extinction risk assessment of range-restricted species. This is even more pressing in the Mediterranean Basin, a global hotspot both for plant diversity and highly threatened taxa [22,23]. In Greece, one of the Mediterranean Basin’s major regional biodiversity and endemism centers [24], and one of the most biodiverse countries in the EU, climate change research is steadily growing, with relevant studies focusing mostly on species-rich, island areas or being taxon-specific [5,20,24,25,26,27,28,29,30,31,32,33,34].

Greece owes its extremely localized endemic flora [20,24] mainly to its rugged topography, the existence of microclimatic refugia, local ecological isolation and rather low climate-change velocity during the Quaternary. However, its rich endemic flora is far from safe, since a large proportion is currently threatened [20]. In this context, extinction risk assessment of its rich endemic flora based on IUCN criteria and standards, provides the baseline for future conservation research and ecosystem services maintenance [20,29]. Additionally, Greece has declared 17 areas as national parks for the protection of wild flora and fauna, for natural ecosystems, as well as for their cultural elements [35,36]. Two of these parks occur in the plant diversity hotspot of the Peloponnese [12], a coastal-wetland type (National Park of Kotychi-Strofilia lagoons), and a mountainous one (Chelmos-Vouraikos National Park). 

Several scientific papers have been published concerning climate change impacts on species distributions and range limits [37,38,39,40,41,42,43] mainly on protected areas [44,45]. Some of those papers have focused on National Parks worldwide (i.e., [46,47,48,49]), as the need for vulnerability information across National Parks is substantial. This is because the climate-driven disturbances with regional characteristics together with regional vegetation or species range shift trends are among the most common indicators for the conservation and management planning of protected areas such as National Parks [37,42,44,46,49]. 

Chelmos-Vouraikos National Park (CVNP) is a floristic diversity and endemism hotspot in Greece [24] and one of the main areas where Greek endemic taxa, assessed as critically endangered and threatened under the IUCN Criteria A and B, are mostly concentrated [20]. The varied relief of the area provides a high habitat and plant taxa diversity and a refuge for many endemic taxa [50]. CVNP is characterized by a high floristic diversity that includes 1467 native plant taxa and 46.26% of the total Peloponnesian vascular flora, as also by high endemism with 177 Greek endemic taxa of which there are 41 Peloponnesian endemics [35]. Endemic plant diversity and its spatial distribution have been studied by [35,50,51] and the presence of five single mountain endemic taxa, namely *Silene conglomeratica* Melzh, *Alchemilla aroanica* (Buser) Rothm., *Lonicera alpigena* subsp. *hellenica* (Boiss.) Kit Tan and Ziel., *Polygala subuniflora* Boiss. and Heldr., and *Valeriana crinii* Boiss. subsp. *crinii*, has been underlined, together with other taxa that occur only at medium to high elevations, such as the Peloponnesian endemics *Corydalis blanda* subsp. *oxelmannii* Lidén and *Globularia stygia* Boiss., that is included in the Annex II of Directive 92/43/EE., as also other rare and range restricted taxa, such as *Bolanthus chelmicus* Phitos subsp. *chelmicus, Campanula aizoides* Greuter, *Cicer graecum* Boiss., *Gymnospermium peloponnesiacum* (Phitos) Strid *and Hieracium greuteri* Gottschl. All these taxa have very small and restricted populations and there is an urgent need for them to be monitored to prevent any negative impact on them.

Conservation strategies are needed to address the ongoing loss of plant diversity, and especially for the protection of rare and threatened plant species [21], even of a single species [52]. The plant micro-reserve (PMR) concept of a less than 20 ha surface protected area aims to protect the main subpopulations of rare endemic taxa through the establishment of a continuous monitoring system [53] and is a complementary tool to preserve biodiversity in larger protected areas [54,55]. Within this context, it is crucial to identify target plant taxa for which monitoring, and conservation are urgent through “regional responsibility” criteria [56,57,58,59,60]. 

In the framework of a project dealing with the development of a plant micro-reserve (PMR) network that will be located on public land, within Chelmos-Vouraikos National Park, the project’s main goal was to design the first PMR network on the Greek mainland, based on previous efforts, experience and outcomes concerning PMRs in Spain, Slovenia, Bulgaria, Cyprus, Greece (on the island of Crete), Italy, and Lebanon [21,53,54,55,61,62,63,64,65,66,67,68,69,70,71,72,73,74]. A designed PMR network includes, among others, populations of local endemic and rare plants, restricted to extremely small areas. Despite the increasing interest on how species occurring in Greece might cope in a hastily warming world and the threat that land cover/land use change poses to biodiversity, no study had ever focused on the effects of climate and land-use change on very narrow mountain endemics, as we proposed here for taxa of the CVNP. To our knowledge, this is the first ever study conducted in Greece and the eastern Mediterranean in general, that incorporates dynamic Land Use Land Cover (LULC) data in a climate change vulnerability assessment (CCVA), as suggested by [75]. CCVAs excluding current and projected LULC data run the risk of underestimating the species future extinction risk [23,75,76]. For these reasons, the main aims of the current study focus on the ecological aspects of the single mountain endemic taxa of CVNP, *Silene conglomeratica*, *Alchemilla aroanica*, *Polygala subuniflora*, and *Valeriana crinii* subsp. *crinii* and three more Peloponnesian endemic taxa, *Corydalis blanda* subsp. *oxelmannii, Globularia stygia* and *Bolanthus chelmicus* subsp. *chelmicus,* that are rare and range restricted, mainly occurring in CVNP. More specifically the aims were (a) to investigate how climate change and land use/land cover change may alter the distribution of these taxa of the National Park via a species distribution modelling approach and, (b) to estimate the current and future extinction risk of the aforementioned taxa based on the IUCN Criteria A and B, in order to investigate the need for designing an effective plant micro-reserve network within the protected areas of CVNP and to support decision making on spatial planning efforts and conservation research for sustainable, integrated management.

## 2. Results

### 2.1. Species Distribution Models

Model performance was exceptional for both model categories, i.e., the climate change (CC) model and the climate change and land-cover/land-use change (CC-LULCC) model; (CC model: AUC: 1.00 ± 0.00; AUC-PR: 0.84 ± 0.02; Brier score: 0.00 ± 0.00; Cohen’s kappa: 1.00 ± 0.00; Somer’s D: 1.00 ± 0.00; Sorensen: 0.8 ± 0.00; TSS: 1.00 ± 0.00; CC- LULCC model: AUC: 1.00 ± 0.00; AUC-PR: 0.80 ± 0.05; Brier score: 0.00 ± 0.00; Cohen’s kappa: 1.00 ± 0.00; Somer’s D: 1.00 ± 0.00; Sorensen: 0.8 ± 0.00; TSS: 1.00 ± 0.00; all values refer to the median; Figure 1; Appendix A; there was no variation between the three different CC- LULCC scenarios). All models performed better than random at *p* < 0.01. All taxa had minimal potential niche truncation values (Appendix A). Variable importance values for each of the taxa included in the analyses are presented in Appendix A. When considering only the dynamic predictors, either the precipitation of the wettest month (PWM) or the temperature annual range (TAR) emerged as the most important variables, in both the CC and the CC-LULCC models (Appendix A). The temporally static variables, i.e., the edaphic and topographical ones may present artificially inflated importance values, as they remain constant in time. This may be logical for the topographical variables in the time-frame we examined, but soil variables are highly likely to be affected by the changing (micro-)climate during the analysed time-span. We focus on the Hadley Centre Global Environment Model version 2 (HadGEM2), Representative Concentration Pathway 8.5 (RCP) combination for the 2080s time-period, as there were insignificant differences between the future projections. The same holds true regarding the CC-LULCC model, so we focus on the Shared Socioeconomic Pathway (SSP) 5, Representative Concentration Pathway 8.5 combination for the same time-period. Extrapolation novelty was minimal across all climate models and time-periods, with the proportion of analogue climate ranged between 95.12–99.98%.

### 2.2. Land Use and Land Cover Changes

In the study area, the extent occupied by broad-leaf deciduous trees (BLDT), needle-leaf evergreen trees (NDLET) and broad-leaf evergreen shrubs (BLES) is expected to increase until 2100 (Figure 2, Appendix A). On the other hand, crop abandonment is projected to be severe (Figure 2, Appendix A) and C3 grasslands are expected to transition rapidly to either barren lands or needle-leaf evergreen trees by 2035, with a relative loss rate almost 5x compared to the mean loss rate among the other LULC classes (Figure 2 and Appendix A). Finally, most of the known localities of the taxa analyzed, fall within areas that are expected to experience 1–3 LULC transition steps (Figure 3).

### 2.3. Habitat Suitability Range Change

Most of the taxa are projected to face range contractions, either in the CC or the CC-LULCC model (Figure 4, Figure 5 and Figure 6). These range contractions are expected to be less intense in the CC-LULCC model (Figure 4, Figure 5 and Figure 6), as the median range contraction is 38.04%. At least one taxon—*Valeriana crinii* subsp. *crinii*—is expected to be stable in the coming decades under the CC-LULCC model (Appendix A). Three taxa, namely *Bolanthus chelmicus* subsp. *chelmicus*, *Corydalis blanda* subsp. *oxelmannii* and *Silene conglomeratica* are projected to experience substantial range decline (over 40%; Appendix A), while the remaining three taxa, i.e., *Alchemilla aroanica*, *Globularia stygia* and *Polygala subuniflora*) are anticipated to lose more than one quarter of their current range (Appendix A). On the other hand, regarding the CC model, all taxa are expected to lose a significant portion of their range, under any time-period and global circulation models (GCM)/RCP combination (Appendix A; Figure 5), with median range contraction being 99.3%. *Alchemilla aroanica* and *Silene conglomeratica* display the highest median range contraction (100%). HadGEM2 and the ensemble GCM showed the lowest and highest mean range contractions, respectively, for any time slice and RCP combination regarding the CC model (Appendix A; Figure 4 and Figure 5), while SSP3-RCP70 and SSP5-RCP85 had the lowest and highest mean range contractions, respectively, regarding the CC-LULCC model (Appendix A; Figure 4 and Figure 6). Figure 7, Figure 8, Figure 9, Figure 10, Figure 11, Figure 12 and Figure 13 present the habitat suitability maps for all the taxa included in the analyses, for both the CC and the CC-LULCC model.

### 2.4. IUCN Extinction Risk Assessment

Most taxa included in our analyses are predicted to continue to be considered as threatened at least inside the CVNP (since three taxa occur in other Peloponnesian mountains as well) based on both Criteria A and B under all GCM/RCP/SSP and time-period combinations in both the CC model and the CC-LULCC model, while *Alchemilla aroanica* and *Silene conglomeratica* are projected to become extinct in most future scenarios under the CC model (Appendix A).

## 3. Discussion

This is the first ever study in Greece and in the eastern Mediterranean in general that has incorporated fine scale dynamic land use/land cover data in a climate change vulnerability assessment in a biodiversity hotspot. CVNP presents an exceptionally high floristic diversity, and it hosts many rare, endemic and threatened species, specialized to different habitats, with distinctive functional traits, requirements and ecosystem services [35,50,51]. 

Our attention is focused on seven rare Peloponnesian and single mountain endemics that are rendered as threatened, as they are habitat specialists that have small population size and a very limited distribution due to their low dispersal ability. It is worth noting that the distribution of most of the species analyzed is mainly governed by static soil factors and by dynamic bioclimatic and LULC variables (Appendix A). The inclusion of high-resolution soil and topographical variables can increase the prediction accuracy of SDMs of very rare plant species [77], as edaphic proxies can discern nuanced variations in their distribution [78,79]. However, the inclusion of static variables may lead to an incomplete representation of future habitat suitability and an artificially inflated importance of such variables in SDMs [80,81,82]. Thus, the variable importance of static predictors should be taken with a pinch of salt regarding future projections. Nevertheless, our models that incorporated both fine-scale static and dynamic explanatory variables, may accurately identify likely locations that are not currently known to host any of the studied taxa. This could guide the efforts of the CNVP management body regarding the discovery of additional unknown populations in these areas or the establishment of an insurance population for the more threatened taxa in a location that is currently uninhabited by the taxon in question, but that is ecologically suitable. After all, soil and topographical characteristics are among the important factors for the distribution of the several endemic and rare taxa occurring in Greece [29]. Most of the local endemic and rare taxa are sensitive to changes in their microhabitat and present a delicate balance with their surrounding environment, a phenomenon observed not only in CVNP, but in other mountain ranges, species complexes and chorological groups in Greece as well [5,26,29,30,32]. Precipitous, calcareous, and often inaccessible cliffs is where the majority of these species dwell and this habitat is characterized by harsh environmental conditions, with a rather stable, low-fluctuating climate (i.e., until now, [83]) and poor soil quality. The distribution of all the studied taxa is co-dominated by a static variable (most often an edaphic one) and a dynamic one, (most often a bioclimatic one; Appendix A). For instance, *Silene conglomeratica* is a local endemic chasmophyte that is specialized to arid conglomerate cliffs. Moreover, *Valeriana crinii* subsp. *crinii,* an obligate chasmophyte and *Globularia stygia,* an occasional chasmophyte, are also dependent on the soil substrate and often the slope and seem to prefer more humid conditions. Most of the studied taxa occupy only a small fraction of their ecologically suitable areas, implying that either stochastic local extinctions have shaped their distribution—as in other rare plant taxa—or barrier effects and subsequent dispersal limitations may have led to limited post-glacial range filling, as observed in other mountain endemics [79,84]. 

Different environmental drivers, climate-related variables and biotic interactions determine the future potential distribution of such rare and locally restricted taxa [85,86,87,88]. To illustrate even further, *Corydalis blanda* subsp. *oxelmannii* prefers limestone substrates and alpine meadows and screes at 1800–2100 m a.s.l. on CVNP, while *Alchemilla aroanica* prefers rather humid areas, near the streams of sparsely vegetated land at the elevational range of 1200–2100 m a.s.l. Furthermore, *Bolanthus chelmicus* subsp. *chelmicus*, and *Valeriana crinii* subsp. *crinii* are usually found at 1200–1800 m a.s.l. and 1200–2000 m a.s.l., respectively, while other taxa, such as *Lonicera alpigena* subsp. *hellenica* which has been found at ca. 1700 m a.s.l., and *Polygala subuniflora* which occurs at 2000–2300 m a.s.l., are found at higher elevations. As species richness is expected to increase with increasing elevation in the near future due to the upward movement of generalist species forced by global warming, this may be followed by the (local) extinction of endemic specialists and rare plants, as a result of a spiral of negative biotic interactions [5,89,90,91,92]. 

Our results showed that all four of the single mountain endemics included in our analyses are projected to continue to be considered as critically endangered and to face large range contractions, under any scenario, at least for the CC model (the anticipated range contractions are less severe under the CC-LULCC model, but still considerable). More specifically, most of those species are predicted to retain their small numbers and limited distribution in the next two decades (2020–2040), but they are projected to become extinct or nearly extinct in the foreseeable future. This indicates that due to their reproduction strategy and life-form, i.e., being chasmophytes, these species might be already experiencing a long-term extinction-debt lag, a phenomenon observed in other long-lived perennials existing in rather unforgiving physical environments [30,93], as well as in other biodiversity hotspots in Greece [5,24]. This could be further exacerbated by the synergistic effects of land-use and land-cover change, which constitute a major concern for most Aegean pollinators as well [27]. Most of the known localities of all the studied taxa occur in areas where LULC changes are anticipated (Figure 3). According to the CC-LULCC model, the expected range contractions are less substantial (Figure 4, Figure 5 and Figure 6; Appendix A). Under this model, one taxon, *Valeriana crinii* subsp. *crinii*, will not experience any change in its range and no taxon is projected to become extinct under the SSPs we analyzed. It seems that the inclusion of LULC change data may ameliorate species endangerment in CVNP, which is not unusual according to a recent meta-analysis [75]. This phenomenon can be attributed—at least for some of the studied taxa—to the expansion of suitable habitat conditions arising from LULC change [75]. This is especially true for *Valeriana crinii* subsp. *crinii*, which occurs mainly in the land cover type of needle-leaf evergreen trees, a type expected to increase its extent in the coming decades in CVNP (Figure 2, Figure 3 and Appendix A). However, *Silene conglomeratica*, one of the rarest Greek endemic taxa and a mountain endemic found exclusively in CVNP, is still expected to face substantial distribution decline (Appendix A). The same holds true for *Bolanthus chelmicus* subsp. *chelmicus* and to a lesser extent for another very rare mountain endemic, *Alchemilla aroanica* (Appendix A). This taxon-specific response to land use/land cover change and climate change is a relatively common phenomenon in the Mediterranean mountains [94,95]. For now, the species populations are quite unaffected by human-induced disturbances, but habitat alteration/degradation and other anthropogenic threats are inescapably expected to escalate in the coming decades especially in the Mediterranean [96], and at a fairly rapid rate even inside a designated protected area such as CVNP (Figure 2, Figure 3 and Appendix A), essentially greatly reducing the species chances of survival, if no conservation actions are taken. The projected afforestation in CVNP, which is in line with previous studies in other Greek mountainous areas (e.g., [97]) may lead to increased diversity of late-successional taxa [98], but at the expense of mountainous grasslands (Figure 2, Figure 3 and Appendix A), where several rare species dwell and hence in reduced species richness in highlands [99]. This means that there exists a very limited time-window for us to implement the actions needed to ascertain their conservation and eventual survival, by applying both in- and ex-situ measures, especially when taking into consideration the rapid LULC change projected in the CVNP during the next 15 years (Appendix A). Two of the rarest plant species occurring in CVNP and in Greece, *Alchemilla aroanica* and *Silene conglomeratica,* are the ones facing the direst prospects, as they are anticipated to become extinct in all time-periods, under any future climate model and scenario, at least in the CC model, indicating that we might need to channel most conservation efforts towards them, such as the deposition of their seeds in seed banks, germplasm collection, mitigation translocation and population reinforcement, thus abiding to Aichi Biodiversity Target 8 (ex-situ conservation of the most threatened plant taxa). Another course of action that should be seriously considered, is the investigation of their population genetic diversity, since genetic data are extremely scant in the Eastern Mediterranean biodiversity hotspot and are warranted if we are to effectively manage and conserve narrow endemic species, their safeguarding of which is pivotal in biological conservation [100]. Ultimately, phenotypic plasticity may mediate—at least for a while—the negative climate-change impacts on extremely localized and genetically depleted taxa [101,102]. Nevertheless, as conservation funds are rather scarce in the unstable post-pandemic economic environment, considering the projection mismatches between the CC and the CC-LULCC models, if only one taxon should be prioritized, then that would have to be *Silene conglomeratica*, as it is the sole taxon that is anticipated to experience very large contractions under any GCM/RCP/SSP combination in both models (i.e., the CC and the CC-LULCC model).

Our predictions could complement future actions, as this is a prerequisite for IUCN assessments and highlights the need for designing an effective plant micro-reserve network within the protected areas of CVNP, a proven cost-effective conservation action [54,56]. Our results also support decision making on spatial planning efforts and conservation research for sustainable management of the populations of such rare, local endemic and range-restricted taxa. Plants with extremely small populations as those used in this study, should be conserved for saving biodiversity, for sustainable development, but also for their evolutionary potential [57]. Assisted colonization and translocation to establish insurance populations could serve as a valuable tool regarding the conservation of very rare and threatened plant taxa [103]. Such an action could be guided by our fine scale SDM that has included dynamic LULC data and high resolution bioclimatic, edaphic and topographical data. 

Regarding future steps, a detailed inventorying of the localities of the target species and their habitats must be implemented, covering population, plant community and habitat type characteristics, their structure and functions and the current pressures and threats they face, to designate and propose precise sites for the plant micro-reserves implementation. Subsequently, monitoring plans should be elaborated by including baseline reference data for every PMR, while conservation targets should be then identified, and the necessary conservation actions proposed. It is very important to use all available information concerning geo- and bio-diversity to prepare and apply effective management plans of the national park [35]. It is essential to designate PMRs of these taxa and then to gain the critical public’s willingness to support them, using one of the most common approaches of providing a park-based education venue [104]. The role of the recently constituted Natural Environment and Climate Change Agency of Greece, is essential to achieve those goals; for the first time all relevant actions are coordinated and funded based on bottom-up identification of each protected areas’ specific requirements and needs, using a top-down governance approach, that guarantees the longevity of conservation and integrated management projects, such as PMR designation and implementation.

## 4. Materials and Methods

### 4.1. Species Occurrence Data

Detailed occurrence data for seven endemic and rare taxa found on CVNP—the four single mountain endemics *Silene conglomeratica*, *Alchemilla aroanica*, *Polygala subuniflora* and *Valeriana crinii* subsp. *crinii* (Figure 14a–d)*,* the regional endemic protected by the Directive 92/43/EE *Globularia stygia*, and the rare and range restricted *Bolanthus chelmicus* subsp. *chelmicus* and *Corydalis blanda* subsp. *oxelmannii*—were obtained from the authors’ field data collected from 2000 and earlier, to 2020. Figure 15 presents the areas where the taxa of interest are found. The single mountain endemic *Lonicera alpigena* subsp. *hellenica* could not be used in the analysis because it is present on only one locality, so it is critically endangered anyway. We removed any points with coordinate uncertainty ≥100 m, as well as eliminating any duplicate points, following [105,106]. Our final occurrence dataset comprised 43 records for the seven selected endemic taxa; all these endemics have at least 3 spatially thinned occurrences, following [107].

### 4.2. Environmental Data

Given that all of our records were collected after 1990, we created baseline monthly climate data for the time-period from 1981 to 2009 and the standard 19 bioclimatic variables presented in WorldClim [108], as well as 16 other environmental variables presented in [109], at a 100 m resolution, using altitudinal data from CGIAR Consortium for Spatial Information [110], working with ClimateEU v4.63, the “dismo” 1.1.4 [111] and the “envirem” 2.2 [109] R packages, following the framework outlined in [112,113,114]. We also incorporated in our analyses soil data derived from SoilGrids [115] and dynamic land cover/land use (LULC) data obtained from [116], which were downscaled to match the resolution of the other abiotic variables using functions from the “raster” 2.6.7 [117] R package. Furthermore, we estimated five topographical variables, namely aspect, heat load index, slope, topographic position index and terrain ruggedness index, based on the same altitudinal data used to construct the environmental variables mentioned earlier, taking advantage of the “raster” 2.6.7 [117] and the “spatialEco” 1.2-0 R packages [118]. 

We generated future climate data for three different time periods (2020s, 2050s and 2080s) and three different models, Community Climate System Model 4 (CCSM4), Hadley Centre Global Environment Model version 2 (HadGEM2), and an ensemble of 15 global circulation models (GCMs) and two different Intergovernmental Panel on Climate Change scenarios from the representative concentration pathways family: RCP4.5 (mild scenario) and RCP8.5 (severe scenario), as described previously. We derived future LULC data from Chen et al. [116], and more specifically, data for the SSP1-RCP26, the SSP3-RCP70 and the SSP5-RCP85 scenarios, where SSP1 and SSP3 constitute the most sustainable and unsustainable path, respectively and SSP5 assumes rapid economic development depending on fossil fuel [119]. Thus, the topographical and soil variables are fixed in time, while the bioclimatic and the LULC variables are temporally dynamic.

Nineteen environmental variables did not present any collinearity issues (Spearman rank correlation < 0.7 and VIF < 5 [120]) and thus were included in the subsequent analyses. Multicollinearity tests were performed with functions from the “usdm” 1.1.18 R package [121]. 

### 4.3. Land Use and Land Cover Changes

Land use and land cover (LULC) and their change (LULCC) are under continuous monitoring since LUCCs have strongly increased over the last decades [122]. We used the OpenLand 1.0.2 R package that provides a comprehensive and integrated suite for the exploratory analysis of LUC changes [122]), as well as to visualize the transition dynamics of LULCC in the study area. We also conducted an intensity analysis with the same R package, based on the LULC data available for the study area, so as to estimate the land cover change rate and the subsequent transition dynamics.

### 4.4. Species Distribution Models

The occurrence to predictors ratio for all seven taxa included in our analyses was lower than 10:1. Consequently, we modeled their realized climatic niche following [123,124,125,126], using functions from the “ecospat” 3.1 R package [127] and taking advantage of the random forest algorithm. We followed [80,128] regarding the creation of pseudo-absences. Occurrence records were divided into 10 different training and testing datasets at an 80 to 20 ratio. Model performance was based on seven different metrics [129,130,131,132,133] and against null models [134].

Well-calibrated (TSS ≥ 0.8) models were used to map the potential suitable area for all taxa for each time-period included in our analyses.

We created binary maps for every possible GCM, RCP and time-period combination by applying the metric that maximizes the sum of sensitivity and specificity [128,135,136]. All non-zero cells in the clamping mask for each taxon were set to NA, as a precautionary measure regarding our predictions [123]. 

The directionality (increase/decrease of their potential suitable area) and extent of the range shift of all seven taxa were assessed with functions from the “biomod2” 3.3.7 R package [137]. All taxa were assumed to have very limited dispersal capacity, which is more appropriate for very range-restricted taxa [138]. We also assessed if there was any niche truncation apparent in our models via the ‘humboldt’ 1.0.0.420121 R package [139]. Finally, we used two metrics (ExDet and the proportion of data nearby) to account for environmental extrapolation via the ‘dsmextra’ 1.1.4 R package [140,141]. The above procedure was repeated for a variable set excluding the dynamic LULC data (the CC model) and for a variable set including the dynamic LULC data (the CC-LULCC model).

### 4.5. Future IUCN Extinction Risk Assessment

Based on the IUCN Criteria A and B, all seven taxa under study were allotted to a preliminary IUCN threat category for each GCM, RCP, SSP and time-period combination based on their distribution inside the CVNP. This was achieved via the “ConR” 1.1.1 R package [142], the R code made available from [41], under the framework applied at a greater spatial extent in Greece by [20]. As a final step, we investigated whether their future and current preliminary IUCN extinction risk status (publicly available from [137]) differed.

## Figures and Tables

**Figure 1 plants-11-03548-f001:**
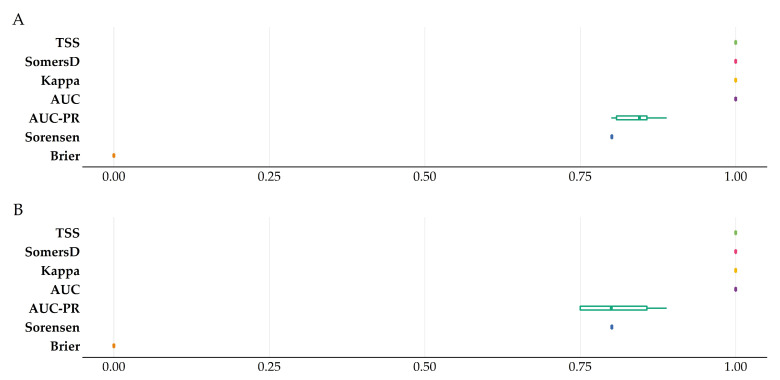
Raincloud plot of the model performance evaluation metrics for (**A**) the CC model and (**B**) the CC-LULCC model. AUC: Area under the curve. AUC-PR: Area under the precision-recall curve. TSS: True skill statistic.

**Figure 2 plants-11-03548-f002:**
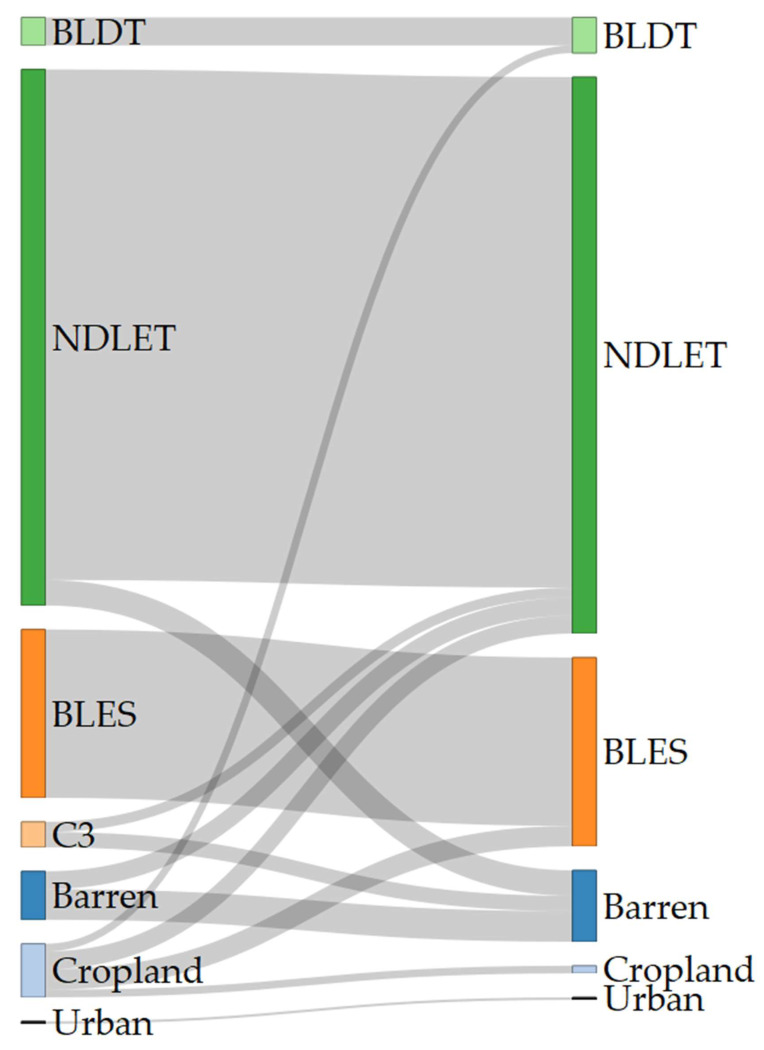
Land cover types and their anticipated transition between 2015 and 2100 in the study area. The left column refers to 2015 and the right column to 2100. Each bar shows the area each land cover type occupies in each year, while the flows between 2015–2100 depict the anticipated transitions between the land cover types. BLDT: broad-leaf deciduous trees. BLES: broad-leaf evergreen shrubs. C3: C3 grasslands. NLDET: needle-leaf evergreen trees.

**Figure 3 plants-11-03548-f003:**
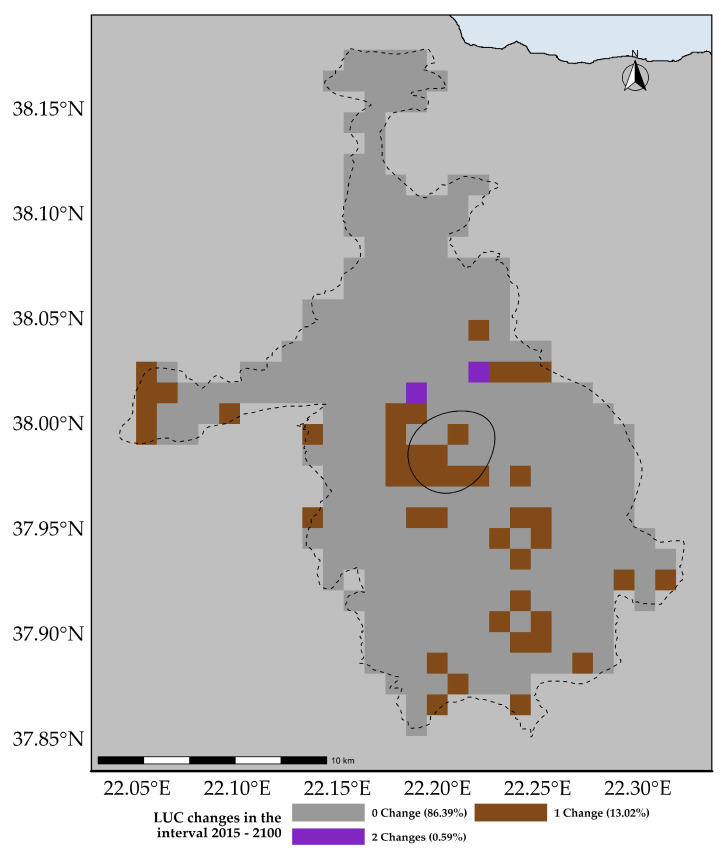
Number of land use and land cover changes anticipated in the study area over the 2015–2100 time-period. The dashed and solid black lines depict the extent of the study area and an ellipse encompassing 75% of the known localities of all the taxa analyzed, respectively.

**Figure 4 plants-11-03548-f004:**
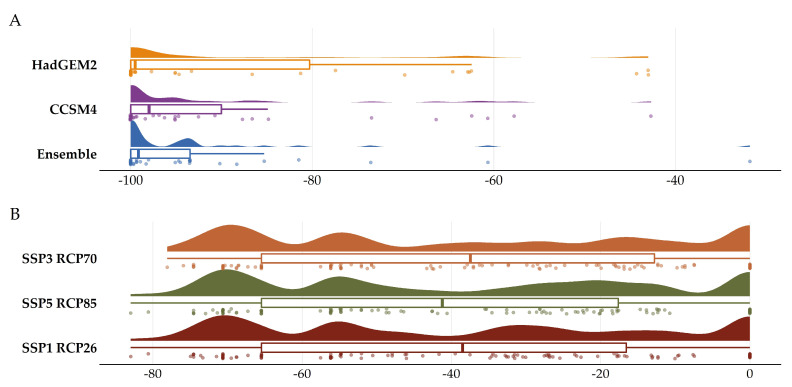
Raincloud plot of the mean projected proportion of area range loss for the seven Greek endemics we analyzed under (**A**) all global circulation models (GCM) under the CC model and (**B**) all shared socioeconomic pathways under the CC-LULCC model.

**Figure 5 plants-11-03548-f005:**
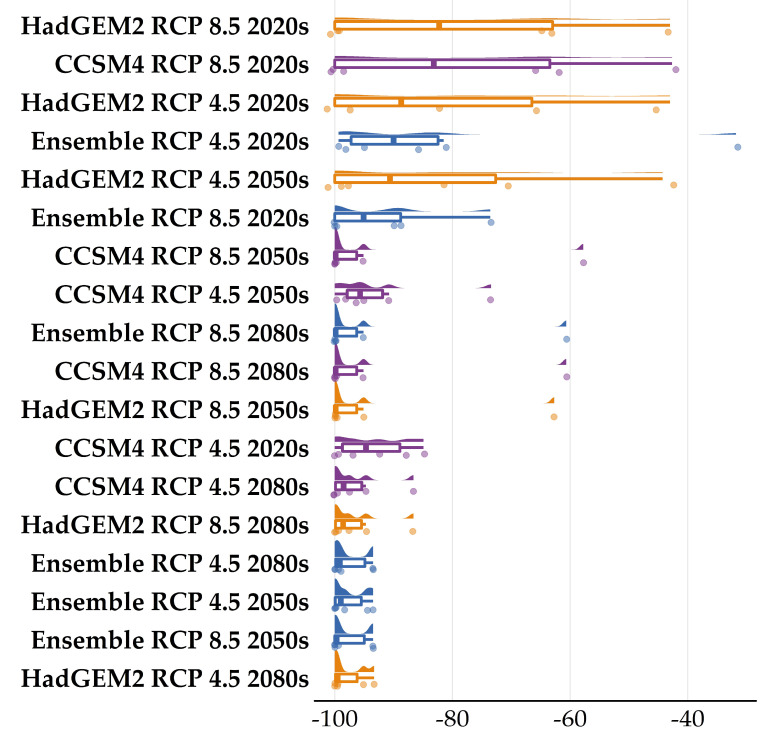
Raincloud plot of the projected proportion of area range loss for the seven Greek endemics we analyzed under all global circulation models (GCM) and representative pathway concentrations (RCP) combination for every time-period under the CC model. The color coding follows Figure 4A.

**Figure 6 plants-11-03548-f006:**
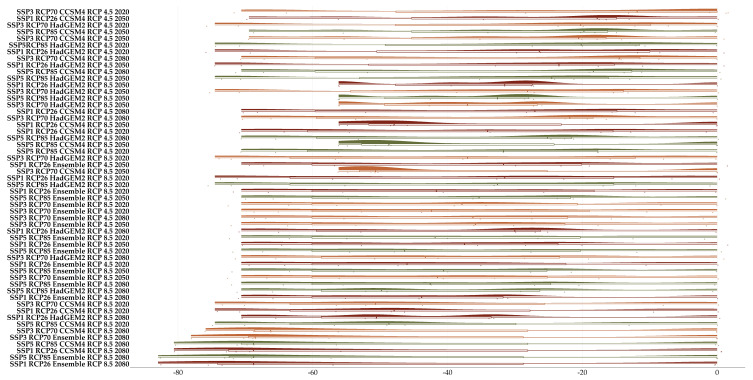
Raincloud plot of the projected proportion of area range loss for the seven Greek endemics we analyzed under all global circulation models (GCM), representative pathway concentrations (RCP) and shared socioeconomic pathways combination for every time-period under the CC-LULCC model. The color coding follows Figure 4B.

**Figure 7 plants-11-03548-f007:**
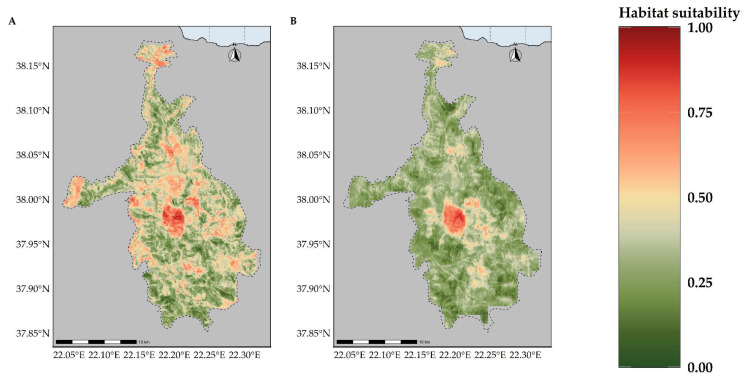
Map of habitat suitability for *Alchemilla aroanica* under the (**A**) CC model and (**B**) CC-LUCCC model. The dashed line denotes the CVNP borders. *Alchelmilla aroanica* is highly specialized regarding its habitat, as it is primarily found near alpine streams and rivulets.

**Figure 8 plants-11-03548-f008:**
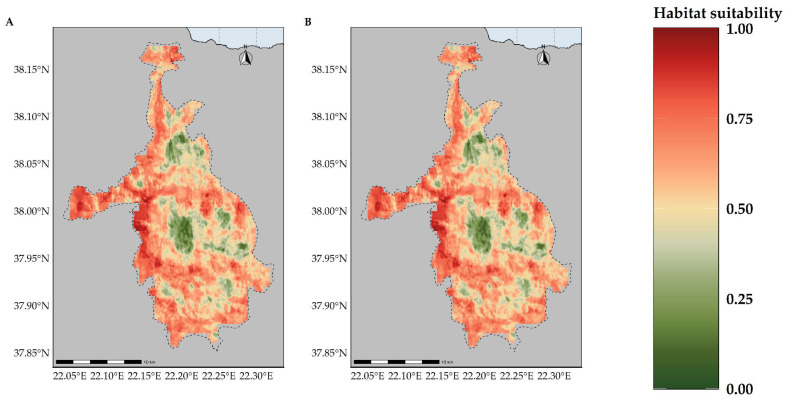
Map of habitat suitability for *Bolanthus chelmicus* subsp. *chelmicus* under the (**A**) CC model and (**B**) CC-LUCCC model. The dashed line denotes the CVNP borders.

**Figure 9 plants-11-03548-f009:**
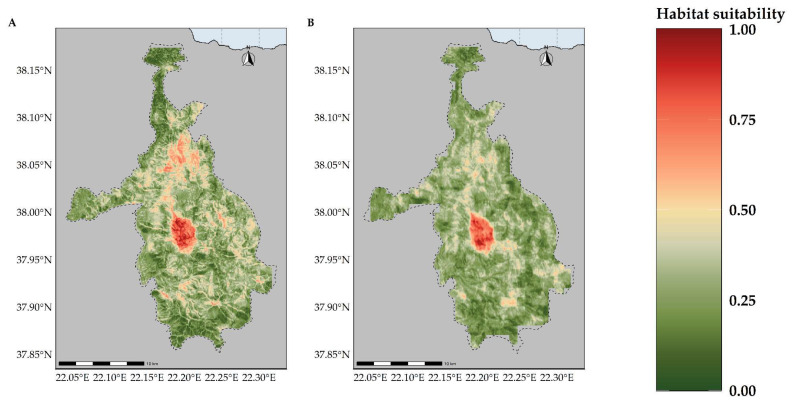
Map of habitat suitability for *Corydalis blanda* subsp. *oxelmannii* under the (**A**) CC model and (**B**) CC-LUCCC model. The dashed line denotes the CVNP borders.

**Figure 10 plants-11-03548-f010:**
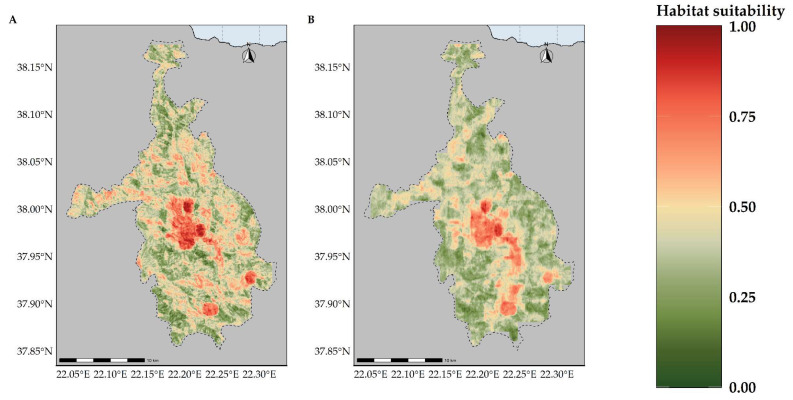
Map of habitat suitability for *Globularia stygia* under the (**A**) CC model and (**B**) CC-LUCCC model. The dashed line denotes the CVNP borders.

**Figure 11 plants-11-03548-f011:**
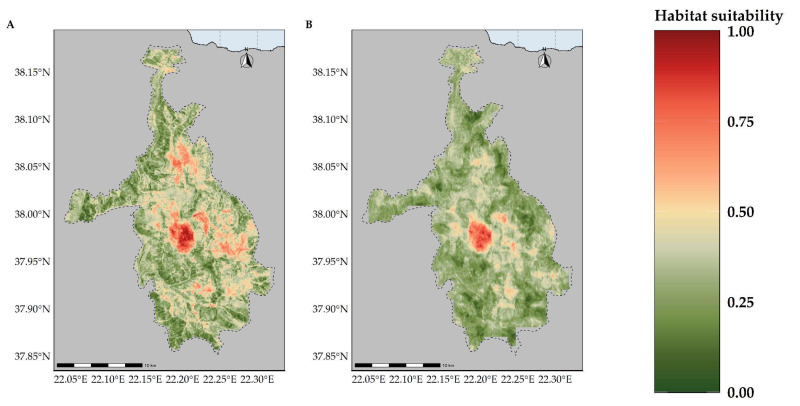
Map of habitat suitability for *Polygala subuniflora* under the (**A**) CC model and (**B**) CC-LUCCC model. The dashed line denotes the CVNP borders.

**Figure 12 plants-11-03548-f012:**
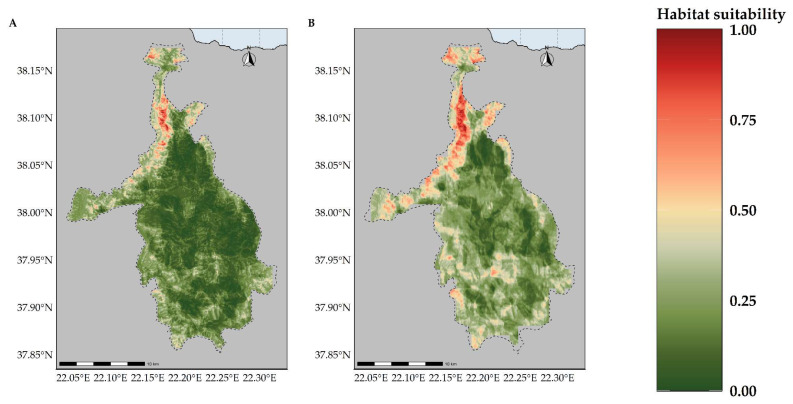
Map of habitat suitability for *Silene conglomeratica* under the (**A**) CC model and (**B**) CC-LUCCC model. The dashed line denotes the CVNP borders.

**Figure 13 plants-11-03548-f013:**
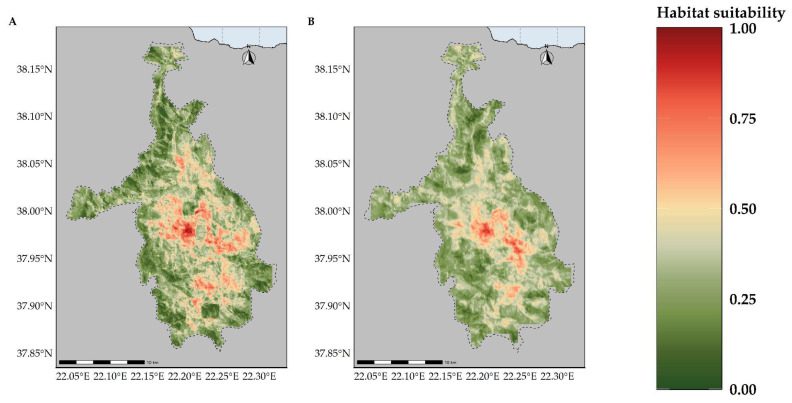
Map of habitat suitability for *Valeriana crinii* subsp. *crinii* under the (**A**) CC model and (**B**) CC-LUCCC model. The dashed line denotes the CVNP borders.

**Figure 14 plants-11-03548-f014:**
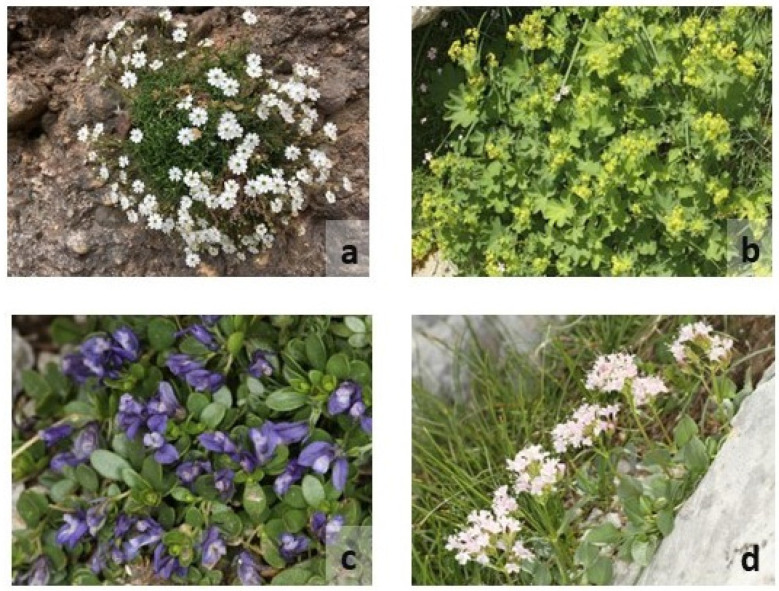
(**a**) *Silene conglomeratica*, (**b**) *Alchemilla aroanica*, (**c**) *Polygala subuniflora* and (**d**) *Valeriana crinii* subsp. *crinii*.

**Figure 15 plants-11-03548-f015:**
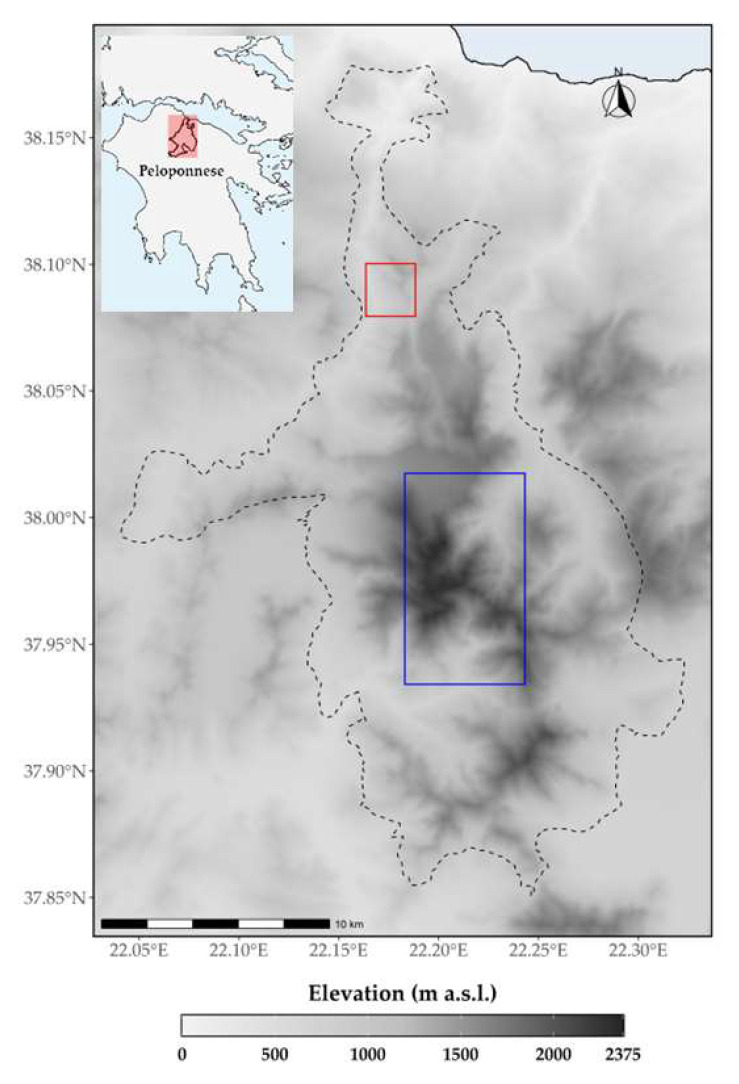
CVNP area map (dashed line). Rectangles represent the areas where the taxa of interest are found. Red: *Silene conglomeratica*; Blue: *Alchemilla aroanica*; *Corydalis blanda* subsp. *oxelmannii*, *Globularia stygia*, *Lonicera alpigena* subsp. *hellenica*, *Polygala subuniflora*, *Valeriana crinii* subsp. *crinii*.

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
