# Peer review of "Climate and Land-Cover Change Impacts and Extinction Risk Assessment of Rare and Threatened Endemic Taxa of Chelmos-Vouraikos National Park (Peloponnese, Greece)"

_plants, 2022, doi:10.3390/plants11243548_

Round 1

Reviewer 1 Report

The manuscript by Kougioumoutzis et al, entitled Climate change impacts and extinction risk assessment of rare and threatened endemic taxa of Chelmos-Vouraikos National Park (Peloponnese, Greece), deals with a topic that is widely debated currently. However, the various aspects of this topic are not treated properly and the article is difficult to read. The sentences are sometimes too long, not well connected and difficult to understand (i.e. lines 141-145; lines 157-163; lines 169-173 etc.). In addition, I think that the research has only a local interest and it is of limited interest for an international audience

In the introduction, it is claimed that the climate has changed since the 1800s mostly due to human activities; although a citation is given, it is not scientifically proven as the causes of climate change are not known and the causes of the current climate change are largely debated.

In the results, the authors should describe the causes of risk for each of the species considered as climate change may not be the unique cause.

In the discussion, little importance is given to other causes of threat such as habitat loss or competition from other species.

The methods are not detailed; they are often not well explained and simply refer to other papers thus requiring a deep knowledge of the specific literature. Furthermore, there are no methodological innovations as already known methods and models are simply applied to a case study.

The data set is based on observations made in a very limited time frame (only 8 years) and they are very low in number (43 records for 7 species).

The images (Figs. 1-2) are of bad quality, almost not legible.

Author Response

Dear Reviewer,

First of all, we would like to thank you for the thoughtful review and the constructive criticism of, and comments on, our manuscript.

Please find below a point-by-point response to your comments:

Point 1: The sentences are sometimes too long, not well connected and difficult to understand (i.e. lines 141-145; lines 157-163; lines 169-173 etc.).

Response 1: Following the reviewer’s suggestion, we have broken one of sentences referred by the reviewer (changes are in italics). We prefer to keep the other two sentences as they stand.

L139-145: ‘Several scientific papers have been published concerning climate change impacts on species distributions and range limits [48–54] mainly on protected areas [55,56]. Some of those papers have focused on National Parks worldwide (i.e., [57–60]), as the need for vulnerability information across National Parks is substantial. This is due to the fact that the climate-driven disturbances with regional characteristics together with the regional vegetation or species range shift trends are among the most common indicators for the conservation and management planning of protected areas as National Parks [48,53,55,57,60].’

Point 2: In addition, I think that the research has only a local interest and it is of limited interest for an international audience

Response 2: We respectfully disagree with the reviewer’s comment. Case studies such as the one we have carried out, can provide valuable input regarding the response of either other equally rare and range-restricted taxa in other parts of the world facing increased aridity and reduced precipitation and might actually pave the way to lay out more systematic, elaborate, climate-sensitive and cost-effective conservation and management plans.

Point 3: In the introduction, it is claimed that the climate has changed since the 1800s mostly due to human activities; although a citation is given, it is not scientifically proven as the causes of climate change are not known and the causes of the current climate change are largely debated.

Response 3: We respectfully disagree with the reviewer. It is widely accepted and can in fact be considered as common knowledge, that indeed recent (in a geological time-frame) climate change is due to human actions starting from the Industrial Revolution and intensifying since the 1950s (the Great Acceleration), a fact acknowledged by IPBES, CBD, IUCN and other international agencies and conventions. We would also like to point out that at least 8 relevant references appear in the first two sentences of our manuscript.

IPBES (2019): Global assessment report on biodiversity and ecosystem services of the Intergovernmental Science-Policy Platform on Biodiversity and Ecosystem Services. E. S. Brondizio, J. Settele, S. Díaz, and H. T. Ngo (editors). IPBES secretariat, Bonn, Germany. 1148 pages. https://doi.org/10.5281/zenodo.3831673

United Nations Climate Change Widespread, Rapid, and Intensifying - IPCC.

Lle Roux, J.J.; Hui, C.; Castillo, M.L.; Iriondo, J.M.; Keet, J.H.; Khapugin, A.A.; Médail, F.; Rejmánek, M.; Theron, G.; Yannelli, F.A.; et al. Recent Anthropogenic Plant Extinctions Differ in Biodiversity Hotspots and Coldspots. Current Biology 2019, 29, doi:10.1016/j.cub.2019.07.063.

Powers, R.P.; Jetz, W. Global Habitat Loss and Extinction Risk of Terrestrial Vertebrates under Future Land-Use-Change Scenarios. Nat Clim Chang 2019, 9.

Li, D.; Olden, J.D.; Lockwood, J.L.; Record, S.; McKinney, M.L.; Baiser, B. Changes in Taxonomic and Phylogenetic Diversity in the Anthropocene. Proceedings of the Royal Society B: Biological Sciences 2020, 287, doi:10.1098/rspb.2020.0777.

Thuiller, W.; Lavergne, S.; Roquet, C.; Boulangeat, I.; Lafourcade, B.; Araujo, M.B. Consequences of Climate Change on the Tree of Life in Europe. Nature 2011, 470, doi:10.1038/nature09705.

Cronk, Q. Plant Extinctions Take Time. Science (1979) 2016, 353, doi:10.1126/science.aag1794.

Nic Lughadha, E.; Bachman, S.P.; Leão, T.C.C.; Forest, F.; Halley, J.M.; Moat, J.; Acedo, C.; Bacon, K.L.; Brewer, R.F.A.; Gâteblé, G.; et al. Extinction Risk and Threats to Plants and Fungi. Plants People Planet 2020, 2.

Cuttelod, A.; García, N.; Abdul Malak, D.; Temple, H.; Katariya, V. The Mediterranean: A Biodiversity Hotspot under Threat. In The 2008 Review of The IUCN Red List of Threatened Species; Vié J.-C., Hilton-Taylor C., Stuart S.N., Eds.; IUCN : Gland, Switzerland, 2008.

Point 4: In the results, the authors should describe the causes of risk for each of the species considered as climate change may not be the unique cause.

Response 4: We respectfully with the reviewer’s suggestion, as the aim of our manuscript as it is explicitly stated both in its title and its aim, is to estimate the extinction risk of these taxa due to climate change and thus it is beyond its scope. After all, these taxa are extremely specialised regarding their habitat and are currently virtually unaffected by other human-induced disturbances.

Point 5: In the discussion, little importance is given to other causes of threat such as habitat loss or competition from other species.

Response 5: We respectfully disagree with the reviewer’s suggestion. As we stated in an earlier response, these taxa are extremely specialised regarding their habitat and are currently virtually unaffected by other human-induced disturbances. Furthermore, we have actually dedicated a large part of the Discussion section on the synergistic effects of climate-change and land-use change on the taxa’s long-term chances of survival and what can be done to tackle this:

L193-222: ‘Our results showed that all four of the single mountain endemics included in our analyses are projected to continue to be considered as Critically Endangered and to face large range contractions, under any scenario. More specifically, most of those species are predicted to retain their small numbers and limited distribution in the next two decades (2020-2040), but they are projected to become extinct or nearly extinct in the foreseeable future. This indicates that due to their reproduction strategy and life-form (i.e., being chasmophytes), these species might be already experiencing a long-term extinction-debt lag, a phenomenon observed in other long-lived perennials existing in a rather unforgiving physical environment [68,83], as well as in other biodiversity hotspots in Greece [5,63]. This could be further exacerbated by the synergistic effects of land-use and land-cover change, which constitute a major concern for most Aegean pollinators as well [66]. For now, the species’ populations are quite unaffected by human-induced disturbances, but habitat degradation and other anthropogenic threats are inescapably expected to escalate in the coming decades, especially in the Mediterranean [84], essentially greatly reducing the species’ chances of survival, if no conservation actions are taken. This means that there exists a very limited time-window for us to implement the actions needed to ascertain their conservation and eventual survival, by applying both in- and ex-situ measures. Two of the rarest plant species occurring in CVNP and in Greece, Alchemilla aroanica and Silene conglomeratica, are the ones facing the direst prospects, as they are anticipated to become extinct in all time-periods, under any future climate model and scenario, indicating that we might need to channel most conservation efforts towards them, such as the deposition of their seeds in seed banks, germplasm collection, mitigation translocation and population reinforcement, thus abiding to Aichi Biodiversity Target 8 (ex-situ conservation of the most threatened plant taxa). Another course of action that should be seriously considered, is the investigation of their population genetic diversity, since genetic data are extremely scant in the Eastern Mediterranean biodiversity hotspot and are warranted, if we are to effectively manage and conserve narrow endemic species, their safeguarding of which is pivotal in biological conservation [85]. Ultimately, phenotypic plasticity may mediate – at least for a while – the negative climate-change impacts on extremely localized and genetically depleted taxa [86,87].’

Point 6: The methods are not detailed; they are often not well explained and simply refer to other papers thus requiring a deep knowledge of the specific literature. Furthermore, there are no methodological innovations as already known methods and models are simply applied to a case study.

Response 6: We respectfully disagree with the reviewer. The Materials and Methods section actually constitutes the second largest part of our manuscript and we thoroughly describe: 1) how we acquired, cleaned and spatially thinned our occurrences and why, 2) how we obtained the environmental data that comprise our predictor variables, their resolution and how we assessed their multicollinearity, 3) how we built and evaluated our models and finally, 4) how we assessed their range shift. Regarding the notion that we simply refer to other papers regarding the methodology used, we would like to note that is the central meaning of publishing scientific knowledge: to disseminate it, make it available to others, back up your work (what/how you did your analyses) and make it easy for everyone interested to check its validity and possibly read other useful papers as well and then apply the same methodology to their data. By doing so, our methodology is as transparent as possible and we make it easy for everyone, even someone who is not so well versed in the SDM analyses, to replicate what we did. As for the other notion, that we did not make any great scientific breakthrough by presenting a novel methodology, well that was not our aim, and quite frankly this is not the aim for the overwhelming majority of researchers working at any field. Case studies on the other hand, can and have been actually proven to be rather useful in exemplifying and showcasing how might some species respond to climate-change (as in this manuscript) or regarding medicine and medical care, have in reality led to elevated interest in rare medical conditions and saved lives. Focusing on our manuscript, investigating how extremely specialised and locally rare plants that do not currently face any other threats might fare in a rapidly changing world in terms of climatic conditions, might actually help us understand how other rare or more widespread taxa might cope with climate change, as these range-restricted taxa may act as beacons of what might other equally rare (e.g., mountain endemics) taxa could be facing in the near future and could thus guide conservation actions, especially when funds are limited.

Point 7: The data set is based on observations made in a very limited time frame (only 8 years) and they are very low in number (43 records for 7 species).

Response 7: Both of these concerns have been already addressed in our manuscript by applying standard SDM methodology, especially created for this type of problems.

Point 8: The images (Figs. 1-2) are of bad quality, almost not legible.

Response 8: This is not fault of our own, but can rather be placed on the publisher’s part, since we have provided high resolution figures. Should the editor decide that these high resolution figures can be shared with the reviewers, this problem would be alleviated.

Reviewer 2 Report

General comments

Reading the paper is sometimes difficult due to its structure. In section 2-Results there are acronyms whose meaning is given in section 3-Materials and methods. Perhaps it would be better to stick to the classic structure (1 Intro, 2 Materials and Methods, 3 Results, …).

Many studies are cited in the Discussion section but there is no comparison between your results and those of other similar studies. If there are no similar studies, it should be declared by emphasizing the originality of the work.

The Discussion section appears too long and partly not focused on the data obtained. With this in mind, the first part could be moved to the Introduction section (see specific comment below).

Specific comments

Line 126. Move "(A)" to the beginning of the sentence.

Line 126 – 128. The resolution of the image is bad in my pdf file, so I can't evaluate it well. However, in caption the meaning of the various elements (colours, abbreviations used) should be explained to allow a better reading.

Line 134. Move "(A)" to the beginning of the sentence.

Line 139 – 163. This part should be moved to the Introduction; in this way the discussion would be more focused on the study data.

Line 166-168. This statement should here be supported by a specific reference to the obtained data.

Line 265. The acronym should be explained in full; at least add “Consortium for Spatial Information”.

Line 275. The acronyms should be explained in full.

Line 310. “Rentacles” change in Rectangles.

Line 315 – 323. Two repeated sentences. Delete one, choose the correct citation (17 or 121) and consequently modify the references.

Line 352. “l” capital letter.

Author Response

Dear Reviewer,

We would like to thank you very much for the thoughtful review and the constructive criticism of, and comments on, our manuscript.

Please find below our point-by-point response to your comments.

Point 1: Reading the paper is sometimes difficult due to its structure. In section 2-Results there are acronyms whose meaning is given in section 3-Materials and methods. Perhaps it would be better to stick to the classic structure (1 Intro, 2 Materials and Methods, 3 Results, …).

Response 1: The reviewer is correct in pointing this out, but this is the journal’s preferred manuscript structure and we simply adhered to it.

Point 2: Line 126. Move "(A)" to the beginning of the sentence.

Response 2: Following the reviewer’s suggestion, we rephrased the sentence in question.

L126: Figure 1. (A) Raincloud plot of the models’ performance evaluation metrics, (B) Raincloud plot of the projected proportion of area range loss for the seven Greek endemics we analysed under all global circulation models (GCM) and representative pathway concentrations (RCP) combination for every time-period.

Point 3: Line 126 – 128. The resolution of the image is bad in my pdf file, so I can't evaluate it well. However, in caption the meaning of the various elements (colours, abbreviations used) should be explained to allow a better reading.

Response 3: The reviewer is correct regarding the image’s resolution, but this is due to the image compression on the journal’s part. We have uploaded high resolution figures according to the journal’s guidelines and these may become available if the editor shares them with the reviewers. Following the reviewer’s comment, we have added the meaning of the abbreviations used in this figure in its legend. The different colours point to the different metrics in panel A and to the different GCM/RCP combinations in panel B.

L126: Figure 1. (A) Raincloud plot of the models’ performance evaluation metrics, (B) Raincloud plot of the projected proportion of area range loss for the seven Greek endemics we analysed under all global circulation models (GCM) and representative pathway concentrations (RCP) combination for every time-period. AUC: Area under the curve. AUC-PR: Area under the precision-recall curve. CBI: Continuous Boyce Index. TSS: True Skill Statistic.

Point 4: Line 134. Move "(A)" to the beginning of the sentence.

Response 4: Following the reviewer’s suggestion, we rephrased the sentence in question.

L134: Figure 2. (A) Map of the elevational gradient of the CVNP and the surrounding area. (B) Map of habitat suitability for Alchemilla aroanica. (C) Map of habitat suitability for Silene conglomeratica. The dashed line denotes the CVNP borders. Alchemilla aroanica is highly specialised regarding its habitat, as it is primarily found near alpine streams and rivulets, while Silene conglomeratica occurs only on conglomerate cliffs.

Point 5: Line 139 – 163. This part should be moved to the Introduction; in this way the discussion would be more focused on the study data.

Response 5: We followed the reviewer’s suggestion.

Point 6: Line 166-168. This statement should here be supported by a specific reference to the obtained data.

Response 6: We have followed the reviewer’s suggestion and added a reference to our results (Table S1).

Point 7: Line 265. The acronym should be explained in full; at least add “Consortium for Spatial Information”.

Response 7: We changed the text according to the reviewer’s suggestion.

L265: ‘…using altitudinal data from CGIAR Consortium for Spatial Information…’

Point 8: Line 275. The acronyms should be explained in full.

Response 8: We changed the text according to the reviewer’s suggestion.

L275: ‘…[Community Climate System Model 4(CCSM4),  Hadley Centre Global Environment Model version 2 (HadGEM2),..’

Point 9: Line 310. “Rentacles” change in Rectangles.

Response 9: Done.

Point 10: Line 315 – 323. Two repeated sentences. Delete one, choose the correct citation (17 or 121) and consequently modify the references.

Response 10: Corrected as suggested.

Point 11: Line 352. “l” capital letter.

Response 11: Done.

Round 2

Reviewer 1 Report

The revised article does not substantially differ from the first version; I understand the authors’ point of view but I remain of my idea.

For this reason, as a reviewer, I believe that the article does not deserve to be published in “Plants” in the present form, but I leave the final decision to the Editor.

Author Response

First of all, we thank the reviewer for the thoughtful review and the constructive criticism of, and comments on, our manuscript during the first revision. Please see below our responses one by one to your comments, for the second revision of our manuscript.

Point 1: The sentences are sometimes too long, not well connected and difficult to understand (i.e. in the first version of the submitted manuscript: lines 141-145; lines 157-163; lines 169-173 etc.).

Response 1: Following the reviewer’s suggestion, we have rephrased these sentences in order to be not very long, clear and well connected. Additionally, lines 141-145 and 157-163 of the first version of the submitted manuscript, have been moved from Discussion to the Introduction.  Lines 169-173 are rephrased and remain at the 2nd paragraph of the Discussion

Point 2: In addition, I think that the research has only a local interest and it is of limited interest for an international audience

Response 2: We believe that case studies such as the one we have carried out, can provide valuable input regarding the response of either other equally rare and range-restricted taxa in other parts of the world facing increased aridity and reduced precipitation and might actually pave the way to lay out more systematic, elaborate, climate-sensitive and cost-effective conservation and management plans.

Point 3: In the introduction, it is claimed that the climate has changed since the 1800s mostly due to human activities; although a citation is given, it is not scientifically proven as the causes of climate change are not known and the causes of the current climate change are largely debated.

Response 3: We have used at least 8 relevant references in the first paragraph of the Introduction concerning this topic. It is widely accepted and can in fact be considered as common knowledge, that indeed recent (in a geological time-frame) climate change is due to human actions starting from the Industrial Revolution and intensifying since the 1950s (the Great Acceleration), a fact acknowledged by IPBES, CBD, IUCN and other international agencies and conventions.

Point 4: In the results, the authors should describe the causes of risk for each of the species considered as climate change may not be the unique cause.

Response 4: These taxa are local endemics of the CVNP, extremely specialised regarding their habitat and are currently virtually unaffected by other human-induced disturbances and for this reason the aim of our manuscript is to estimate the extinction risk of these taxa due to climate change.

Point 5: In the discussion, little importance is given to other causes of threat such as habitat loss or competition from other species.

Response 5: As we stated in Response 4, these taxa are extremely specialised regarding their habitat and are currently virtually unaffected by other human-induced disturbances. Furthermore, we have dedicated a large part of the Discussion section on the synergistic effects of climate-change and land-use change on the taxa’s long-term chances of survival and what can be done to tackle this.

Point 6: The methods are not detailed; they are often not well explained and simply refer to other papers thus requiring a deep knowledge of the specific literature. Furthermore, there are no methodological innovations as already known methods and models are simply applied to a case study.

Response 6: The Materials and Methods section actually constitutes the second largest part of our manuscript and we thoroughly describe: 1) how we acquired, cleaned and spatially thinned our occurrences and why, 2) how we obtained the environmental data that comprise our predictor variables, their resolution and how we assessed their multicollinearity, 3) how we built and evaluated our models and finally, 4) how we assessed their range shift. Regarding the notion that we simply refer to other papers regarding the methodology used, we would like to note that is the central meaning of publishing scientific knowledge: to disseminate it, make it available to others, back up your work (what/how you did your analyses) and make it easy for everyone interested to check its validity and possibly read other useful papers as well and then apply the same methodology to their data. By doing so, our methodology is as transparent as possible and we make it easy for everyone, even someone who is not so well versed in the SDM analyses, to replicate what we did. As for the other notion, that we did not make any great scientific breakthrough by presenting a novel methodology, well that was not our aim, and quite frankly this is not the aim for the overwhelming majority of researchers working at any field. Case studies on the other hand, can and have been actually proven to be rather useful in exemplifying and showcasing how might some species respond to climate-change (as in this manuscript) or regarding medicine and medical care, have in reality led to elevated interest in rare medical conditions and saved lives. Focusing on our manuscript, investigating how extremely specialised and locally rare plants that do not currently face any other threats might fare in a rapidly changing world in terms of climatic conditions, might actually help us understand how other rare or more widespread taxa might cope with climate change, as these range-restricted taxa may act as beacons of what might other equally rare (e.g., mountain endemics) taxa could be facing in the near future and could thus guide conservation actions, especially when funds are limited.

Point 7: The data set is based on observations made in a very limited time frame (only 8 years) and they are very low in number (43 records for 7 species).

Response 7: Both of these concerns have been already addressed in our manuscript by applying standard SDM methodology, especially created for this type of problems. Additionally we would like to mention that field work is continuous on Chelmos – Vouraikos area for more than 40 years for two of the authors (D.T. and G.I.), two authors have worked on the area for their PhD thesis (M.T and E.K), E.K is the Director of the Management Unit of K.K, P.T and I.P.K also visit CVNP for field work for more than 10 years. So, the authors know very well the plant species composition of CVNP and the pressures and threats on the flora. The taxa used in the analysis are very rare and the 43 records represent the different locations of the 7 taxa and number of individuals registered there.

Point 8: The images (Figs. 1-2) are of bad quality, almost not legible.

Response 8: We changed the images in this version although we had provided them in high resolution to the previous submissions, too. Should the editor decide that these high resolution figures can be shared with the reviewers, this problem would be alleviated.